# AN INVARIANT INFORMATION GEOMETRIC METHOD FOR HIGH-DIMENSIONAL ONLINE OPTIMIZATION

## ABSTRACT

Sample efficiency lies at the heart of many optimization problems, especially for black-box settings where costly evaluations and zeroth-order feedback occur. Typical methods such as Bayesian optimization and evolutionary strategy, which stem from an online formulation that optimizes mostly through the current batch, suffer from either high computational cost or low efficiency. To strengthen sample efficiency under reasonable computational cost, one promising way is to achieve invariant under smooth bijective transformations of model parameters. In this paper, we build the first invariant practical optimizer framework INVIGO based on information geometric optimization. It can incorporate historical information without violating the invariant. We further exemplify INVIGO with historical information on multi-dimensional Gaussian, which gives an invariant and scalable optimizer SYNCMA that fully incorporates historical information with no external learning rate to tune. The theoretical behavior and advantages of our algorithm over other Gaussian-based optimizers are further analyzed to demonstrate its theoretical superiority. We then benchmark SYNCMA against other leading optimizers, including the competitive optimizer in Bayesian optimization, on synthetic functions, Mujoco locomotion tasks and rover planning task. In all scenarios, SYNCMA demonstrates great competence, if not dominance, over other optimizers in sample efficiency.

## 1 INTRODUCTION

For many optimization problems in the real world, we do not have access to any gradient information in the continuous space. The available zeroth-order evaluations, on the other hand, are so expensive to collect that their usefulness may decrease tremendously with time as the environment changes. Therefore, it may be better to model these tasks as online optimization problems with zeroth-order feedback and an ignorant initial. And an ideal optimizer should have high sample efficiency with reasonable computational complexity.

Typical methods in this scenario are mostly from the field of black-box optimization, such as Bayesian optimization and evolutionary strategy. Both of them give more weight to the current information to either adapt a global surrogate model or performs a local search strategy, so they also thrive in the online settings. In the previous literature, these two methods may use different names and criteria under similar formulations, so we compare our method with Bayesian optimization and evolutionary strategy under the same criteria as shown in the Experiments section.

Bayesian optimization has great sample efficiency but limited scalability. It suffers from cubic or quadratic computational complexity with respect to sample size, which severely limits its use when the dimension is beyond a few dozen. There exist scalable variants such as TuRBO (Eriksson et al., 2019), but their computational cost is still order of magnitude higher than evolutionary strategies.

Evolutionary strategies include a number of nature-inspired optimizers, among which the leading family of optimizers is originated from covariance matrix adaptation evolutionary strategy(CMA-ES) (Hansen, 2016). These CMA optimizers, such as DD-CMA (Akimoto & Hansen, 2020) and TR-CMA-ES (Abdolmaleki et al., 2017), choose multi-dimensional Gaussian as their parameter space which we will exemplify. CMA optimizers reach a balance between sample efficiency and computational cost, but lack a solid theoretical foundation for interpretation and analysis. Therefore, despite many efforts invested (Arnold & Hansen, 2010; Brockhoff et al., 2012; Shirakawa et al.,

2018; Nishida & Akimoto, 2018; Ba et al., 2016; Akimoto & Hansen, 2016), limited improvement in sample efficiency has been achieved.

For general optimization problems where gradient information is available, the performance of leading first-order optimizers such as AdaGrad (Duchi et al., 2011) and Adam (Kingma & Ba, 2014) are highly dependent on the curvature of the optimization objective. Since the curvature depends on the parameterization of the model, reparameterization techniques (Salimans & Kingma, 2016; He et al., 2016) or a parameterization invariant optimizer are considered promising ways to optimize high-value or even life-critical systems that have rugged objectives. For invariant optimizers, natural gradient (Amari, 1998) uses the local landscape of the parameter space to be invariant in an ideal case. Further efforts (Song et al., 2018; Transtrum & Sethna, 2012) towards practical methods then concentrate on exploiting higher order structure in parameter space, i.e. geodesic in statistical manifold, to accelerate or strengthen invariant for natural gradient. However, only limited progress has been made.

When gradient information is not available, the primary mission for invariant is to compute the gradient and local curvature with sampling. Information Geometric Optimization(IGO) (Ollivier et al., 2017) makes a solid step forward for parametric distributions, allowing limited natural gradient with only zeroth-order feedback. It encompasses several evolutionary strategies and classical statistical models in a unified framework. Similarly, geodesic modification is also explored (Bensadon, 2015) but the practical invariant capability is limited as in the general case.

In this paper, we build the first invariant optimizer framework INVIGO for online optimization with ignorant initial and zeroth-order feedback. INVIGO adopts an approximation to the objective in IGO to allow everywhere differentiability and no external learning rate. While this approximation can be solved directly with duality, we use only Lagrange necessary condition to construct the solution in order to leave room for complete and scalable incorporation of historical information. The INVIGO with historical information is built thereby with an invariant property, which is further exemplified with multi-dimensional Gaussian to derive the practical optimizer SYNCMA. SYNCMAis a synchronous optimizer that is invariant, scalable, free of external learning rate, and historical information completely incorporated. We then analyze its theoretical advantages and connection over other optimizers. Empirically, in both synthetic and realistic tasks, SYNCMA demonstrates great competence over other optimizers in sample efficiency.

To summarize, our main contribution is the invariant framework INVIGO which exemplifies an invariant, scalable, external learning rate free and historical information fully incorporated optimizer SYNCMA with edges in synthetic and realistic tasks against other optimizers both theoretically and empirically.

## 2 AN INVARIANT OPTIMIZER FAMILY WITH AN APPROXIMATE OBJECTIVE

Considering the online optimization problem where a black-box function $f$ needs to be optimized, and the optimizer is initially ignorant with only zeroth-order feedback available.

$$x^* = \mathrm{argmin}_{x \in \mathbb{R}^n} \ f(x) \tag{1}$$

A global parametric sample distribution $\theta \mapsto p_\theta$ is often used to relax the original optimization problem into an optimization problem on the parameter space $\Theta$.

$$\theta^* = \mathrm{argmin}_{\theta \in \Theta} \ \mathbb{E}_{p_\theta}[f(x)] \tag{2}$$

However, as $f$ itself might not be continuous or finite, typical methods may use an substitutional fitness function $g_{f,\theta}(x)$ to represent how good a sample $x \in \mathbb{R}^n$ is.

$$\theta^* = \mathrm{argmin}_\theta \ \mathbb{E}_{p_\theta}[g_{f,\theta}(x)] \tag{3}$$

The expression of $g$ is determined manually from the set of integrable functions depending on original objective $f$ and perhaps the current point $\theta \in \Theta$. The substitution here actually generalize the problem as specifically, (1) If $f$ is good, then it is naturally to set $g_{f,\theta} = f$; (2) If $g$ is related with $\theta$, then the problem will match the time-varying environment setting for which lots of online optimizers pursue.

## 2.1 Natural Gradient Flow with Zeroth-order Feedback

To solve equation (3) on the parameter space $\Theta$, a natural gradient flow is often the primary choice. Here $\theta^t$ denotes the current $\theta$ with static nature that gradient should not apply on.

$$\frac{\mathrm{d}\theta}{\mathrm{d}t} = -\tilde{\nabla}_\theta \mathbb{E}_{p_\theta}[g_{f,\theta^t}(x)] \tag{4}$$

The usage of natural gradient keep this ODE invariant under smooth bijective transformation of parameter space. Its vanilla discrete version, i.e. natural gradient descend algorithm goes to,

$$\theta^{t+1} = \theta^t - h\tilde{\nabla}_\theta|_{\theta=\theta^t} L_{\theta^t}(\theta) \tag{5}$$

Here $h$ denotes the learning rate, and the loss function is defined as $L_{\theta^t}(\theta) \equiv \mathbb{E}_{p_\theta}[g_{f,\theta^t}(x)]$. This definition of loss function can be easily extended to contain the usual definition of loss function in deep learning by extending the sample distribution $p_\theta$ to include neural network, although in this paper we concentrate on parametric distribution families.

To practically compute the natural gradient from ignorant initial and zeroth-order feedback, IGO stems from the fact that $\tilde{\nabla}_\theta p_\theta = p_\theta \tilde{\nabla}_\theta \ln p_\theta$ when $p_\theta$ is smooth. They discover for certain distribution families such as multi-dimension Gaussian, it is possible to compute natural gradient at current point $\theta^t$ through sampling. Here $I(\theta)$ denotes the Fisher information matrix of $p_\theta$, indicating the local landscape of parameter space.

$$\tilde{\nabla}_\theta|_{\theta=\theta^t} L_{\theta^t}(\theta) = \tilde{\nabla}_\theta|_{\theta=\theta^t} \int g_{f,\theta^t}(x) p_\theta(x) \mathrm{d}x \tag{6}$$

$$= I^{-1}(\theta^t) \int g_{f,\theta^t}(x) \frac{\partial \ln p_\theta(x)}{\partial \theta}|_{\theta=\theta^t} p_{\theta^t}(\mathrm{d}x) \tag{7}$$

In IGO, the natural gradient is only available at point $\theta^t$, different choices of distribution family $\Theta$ thus give different optimization methods in the form of (5). In other words, certain $\Theta$ must satisfy the following assumption in order to apply IGO. Also as currently there is no alternative to IGO, we define the IGO complexity to measure the computational complexity of other natural gradient based optimizers.

**Assumption 1.** *For a given $x \in \mathbb{R}^n$ and $\theta \in \Theta$, $I^{-1}(\theta)\frac{\partial \ln p_\theta(x)}{\partial \theta}$ cost finite $H$ time to compute.*

**Definition 1** (IGO complexity). *When assumption1 holds, the IGO complexity $\mathcal{O}(HN)$ is the computational complexity for single step updates when applying IGO to natural gradient method, i.e. to compute equation (7) with $N$ samples.*

## 2.2 Invariant Error in Current Methods

When discretizing with certain learning rate, errors with respect to the invariant property will occur. Here we adopt the definition in Song et al. (2018) to characterize invariant error with invariant order. We say that an optimizer is $d$-th order invariant if the error between the approximate solution and some exactly invariant trajectory decreases as $O(h^d)$.

Unfortunately, in general when the natural gradient is accessible, there is no known complete invariant optimizer for an arbitrary $\Theta$ under computational complexity comparable to natural gradient descend algorithm. The best so far result is 2nd order invariant. In the context of black-box objective with zeroth-order feedback, where the natural gradient is not intrinsic available, IGO under assumption 1 gives accessibility to natural gradient descend and thus is 1st order invariant. All these results are far from complete invariant.

**Definition 2** (Invariant property). *Let $\theta$ be the parameter of an optimizer using model $p_\theta$ and $\varphi(\theta)$ be an smooth bijective transformation of $\theta$ of the same optimizer using model $p'_{\varphi(\theta)} = p_\theta$. Let $\theta^t$ be the optimization trajectory when optimizing objective $f$, parameterized by $\theta$ and initialized at $\theta^0$. And $\varphi^t$ the optimization trajectory when optimizing objective $f$, parameterized by $\varphi$ and initialized at $\varphi^0 = \varphi(\theta^0)$. We say this optimizer is invariant if $\forall t \in \mathbb{N}, \varphi^t = \varphi(\theta^t)$.*

## 2.3 Optimizing with the Approximate Objective : InvIGO

Given that $g_{f,\theta^t}$ is manually selected and $\forall b \in \mathbb{R}, \nabla_\theta E_{p_\theta}[g_{f,\theta^t}(x)] = \nabla_\theta E_{p_\theta}[g_{f,\theta^t}(x) + b]$, we assume $g_{f,\theta^t}$ to be non-negative without loss of generality. Let reweighted distribution $q_\theta(x) \equiv$

$\frac{p_\theta(x)g_{f,\theta^t}(x)}{L_{\theta^t}(\theta)}$, then we can decompose $\log L_{\theta^t}(\theta)$ as follow,

$$\log \frac{L_{\theta^t}(\theta)}{L_{\theta^t}(\theta^t)} = \int q_{\theta^t}(x)(\log \frac{q_{\theta^t}(x)}{q_\theta(x)} + \log \frac{p_\theta(x)}{p_{\theta^t}(x)})\mathrm{d}x \tag{8}$$

$$= D_{KL}(q_{\theta^t}||q_\theta) + D_{KL}(q_{\theta^t}||p_{\theta^t}) - D_{KL}(q_{\theta^t}||p_\theta) \tag{9}$$

Inspired from this decomposition, we claim $D_{KL}(q_{\theta^t}||p_\theta)$ a good objective approximating $L_{\theta^t}(\theta)$. All the proofs and detailed derivations are in Appendix A.

**Theorem 1.** *The KL-divergence $D_{KL}(q_{\theta^t}||p_\theta)$ is a substitution for $L_{\theta^t}(\theta)$ with the following properties.*

1. *The (natural) gradients for $\log L_{\theta^t}(\theta)$ and $-D_{KL}(q_{\theta^t}||p_\theta)$ coincide at current point $\theta^t$, further $\forall \theta \equiv (\theta^t + \delta\theta) \in \Theta$, $\nabla_\theta \log L_{\theta^t}(\theta) = -\nabla_\theta D_{KL}(q_{\theta^t}||p_\theta) + O(\delta\theta)$.*

2. *Under Assumption 1 , computing natural gradient of $D_{KL}(q_{\theta^t}||p_\theta)$ at any point $\theta \in \Theta$ costs the IGO complexity $O(HN)$. While objective $L_{\theta^t}(\theta)$ in IGO is only available to be differentiated at point $\theta^t$.*

Combining with above two properties of $D_{KL}(q_{\theta^t}||p_\theta)$, it is natural to consider a step size constraint update for $\theta^{t+1}$ when optimizing $D_{KL}(q_{\theta^t}||p_\theta)$. The specific choice of that step size constraint comes from the definition of natural gradient,

$$\tilde{\nabla}|_{\theta=\theta^t} L_{\theta^t}(\theta) = -L_{\theta^t}(\theta^t)\tilde{\nabla}|_{\theta=\theta^t} D_{KL}(q_{\theta^t}||p_\theta) \tag{10}$$

$$\propto \lim_{\epsilon \to 0^+} \frac{1}{\epsilon} \mathrm{argmax}_{\delta\theta \ s.t. \ D_{KL}(p_{\theta^t}||p_{\theta^t+\delta\theta}) \le \epsilon^2/2} D_{KL}(q_{\theta^t}||p_{\theta^t+\delta\theta}) \tag{11}$$

Now our optimization problem for each time step is,

$$\theta_*^{t+1} = \mathrm{argmax}_\theta D_{KL}(q_{\theta^t}||p_\theta) \tag{12}$$

$$s.t. \ D_{KL}(p_{\theta^t}||p_\theta) \le \epsilon^2/2$$

While formulation (12) is similar to the basic problem formulation of TR-CMA-ES, it is clear that our invariant motivation gives a different road to (12) and further leads to a different modification and final framework. Specifically, TR-CMA-ES follows Expectation-Maximazation framework to formulate and thus directly solves it with strong duality and an additional search, and arbitrarily uses historical information. As our motivation is invariant under proper computational cost, we only apply natural Lagrange condition over multiplier $\eta$ to derive our updates that enables further scalably and completely incorporating historical information.

$$\tilde{\nabla}_\theta|_{\theta=\theta^{t+1}}(-D_{KL}(q_{\theta^t}||p_\theta) + \eta(\epsilon^2/2 - D_{KL}(p_{\theta^t}||p_\theta))) = 0 \tag{13}$$

We name such algorithm family from iteratively solving (13) for different choice of parametric distribution family $\Theta$ as INVIGO.

**Assumption 2.** *The chosen fitness function $g_{f,\theta^t}(x)$ and the Lagrange multiplier $\eta$ are independent from the parameterization of $\theta$.*

**Theorem 2** (Invariant for INVIGO). *When assumption 1 and 2 hold, optimizers in INVIGO are invariant. While the single step computational cost remains the same as IGO complexity $\mathcal{O}(HN)$.*

Apart from the desired invariant property, one may ask what is the behavior when optimizing problem (12), as equation (13) is just the necessary condition of optimization problem (12). According to equation (11), it can be seen that the optimization problem (12) is solving ODE :

$$\frac{\mathrm{d}\theta}{\mathrm{d}t} = -s(\theta)\tilde{\nabla}\log L_{\theta^t}(\theta) \tag{14}$$

Here $s(\theta) \equiv \frac{1}{\epsilon}||\tilde{\nabla}|_{\theta=\theta^t} D_{KL}(q_{\theta^t}||p_\theta)||$ corresponds to the implicit learning rate of INVIGO. As there is no explicit learning rate, this implicit learning rate hints an overall learning rate that is proportion to $D_{KL}(p_{\theta^t}||p_\theta)^{-0.5}$. The efficiency of such dependency of the overall learning rate with the KL-divergence between adjacent distributions is widely verified in natural gradient based optimization methods such as K-FAC Ba et al. (2016) and reinforcement learning algorithms such as ACKTR Wu et al. (2017). Our formulation in constraint optimization problem 12 is justified thereby.

Additionally, the efficiency of the overall learning rate without an explicit learning rate demonstrate the decrease in the number of hyperparameters to tune with. When further exemplifying with specific distribution family and comparing with other mature algorithms in online optimization, this decrease will be our advantage as shown in the following exemplification.

## 2.4 INVARIANTLY INCORPORATING HISTORICAL INFORMATION

When only local information is used in each iteration, historical information is useful for the optimization, even if the environment is continuously changing over time. We thus demonstrate how to incorporate historical information in INVIGO without violating the invariant property. The modification is changing the objective $D_{KL}(q_{\theta^t} || p_\theta)$ to incorporate historical information. Here $T$ denotes the horizon and the widely used exponential decay is applied with decay parameter $\lambda \in [0, 1)$, $\lambda^0$ is regarded as 1 in default.

$$\theta_*^{t+1} = \text{argmax}_\theta \sum_{\tau=0}^{T} \lambda^\tau D_{KL}(q_{\theta^{t-\tau}} || p_\theta) \tag{15}$$

$$s.t. \ D_{KL}(p_{\theta^t} || p_\theta) \le \epsilon^2/2$$

Similar to equation (13), applying natural Lagrange condition on formulation (15) yields corresponding update. We name the algorithm family from iteratively following that updates for different choice of parametric distribution family $\Theta$ as INVIGO with historical information.

$$\tilde{\nabla}_\theta|_{\theta=\theta^{t+1}} (-\sum_{\tau=0}^{T} \lambda^\tau D_{KL}(q_{\theta^{t-\tau}} || p_\theta) + \eta(\epsilon^2/2 - D_{KL}(p_{\theta^t} || p_\theta))) = 0 \tag{16}$$

**Theorem 3** (Invariant for INVIGO with historical information). *When assumption 1, 2 hold and the decay weight $\lambda$ is independent from the parameterization of $\theta$, optimizers in INVIGO with historical information are invariant. With the single step computational cost $\mathcal{O}(THN)$ in general.*

In some parametric distribution family, it is possible to represent $\tilde{\nabla}_\theta \sum_{\tau=0}^{T} \lambda^\tau D_{KL}(q_{\theta^{t-\tau}} || p_\theta)$ as a self-evolved term to decrease the computational cost to $\mathcal{O}(HN)$ as shown in our following exemplification.

## 3 EXEMPLIFYING WITH MULTI-DIMENSIONAL GAUSSIAN

We choose multi-dimensional Gaussian as our candidate distribution family $\Theta$ given its wide applications.

To start with, we claim the computational accessibility of multi-dimensional Gaussian for assumption 1,

**Proposition 1** (Theorem 4.1 in Akimoto et al. (2012)). *Suppose $\theta_m$ and $\theta_c$ are $n$- and $n(n + 1)/2$-dimensional column representing mean and covariance respectively. Then $\partial m/\partial \theta_m$ and $\partial vec(C)/\partial \theta_c$ are invertible at $\theta \in \Theta$ and,*

$$I_m^{-1}(\theta) \frac{\partial \ln P_\theta(x)}{\partial \theta_m} = (\frac{\partial m}{\partial \theta_m})^{-1}(x - m) \tag{17}$$

$$I_c^{-1}(\theta) \frac{\partial \ln P_\theta(x)}{\partial \theta_c} = (\frac{\partial vec(C)}{\partial \theta_c})^{-1} vec((x - m)(x - m)^T - C) \tag{18}$$

Then we propose our choices of the fitness function $g_{f,\theta^t}(x)$ and the Lagrange multiplier $\eta$ to satisfy assumption 2 while being as simple as possible. We denote the fitness function $g_{f,\theta^t}(x)$ as the level function that reflect the probability to sample a better value from $p_{\theta^t}$. This is the exact choice used in standard CMA-ES Hansen (2016) . In time step $t$, $N$ samples $\{x_i^t\}$ are drawn from $p_{\theta^t}$ and we further denote $\hat{w}_i^t \equiv \frac{g_{f,\theta^t}(x_i^t)}{\sum_i g_{f,\theta^t}(x_i^t)}$ as the normalized fitness for sample $x_i^t$.

According to proposition 1, parameter $\theta = (\theta_m, \theta_c) \mapsto \mathcal{N}(m, C)$ with $\theta_m \in \mathbb{R}^n$ and $\theta_c \in \mathbb{R}^{n(n+1)/2}$ representing mean and covariance respectively. We can thus split the Lagrange multiplier into $\eta = (\eta_m, \eta_c)$ in INVIGO without violating the invariant property. For better comparisons with CMA family optimizers, we adopt this split to directly use the default values in the fine-tuned version of CMA-ES and keep them constant. The assumption 2 is satisfied therefore.

We use the parameterization $\theta = (m, C)$ for simplification sake through this section. Different parameterizations that meet the conditions in proposition 1 will conduct different practical optimizers

by following this section with minor modifications. The performance should be the same up to the transformation due to the invariant property but may vary if certain modification that violates the conditions in theorem 3 is made for other needs.

## 3.1 An Invariant Optimizer with Historical Information : SynCMA

We directly apply InvIGO with historical information and a maximum time horizon $T = t - 1$, the objective is thereby abbreviated as $G^t(\theta) \equiv \sum_{\tau=0}^{T} \lambda^\tau D_{KL}(q_{\theta^{t-\tau}}||p_\theta)$. By choosing such infinite horizon, the historical information is maximally used with the price of $t$ times computational costs more than IGO. This proportion to the current sample size is exactly the defect of Bayesian optimization. Fortunately, we can overcome this extra proportion and reduce the computational costs to the same as IGO, i.e. scalable to $\mathcal{O}(HN)$ for single step update.

The key is to replace $G^t(\theta)$ with a self-evolved term $M^t(\theta)$, such that the gradient information is completely preserved.

$$\tilde{\nabla}_\theta G^t(\theta) = -\tilde{\nabla}_\theta M^t(\theta) + \tilde{\nabla}_\theta D_{KL}(q_{\theta^t}||p_\theta) \tag{19}$$

As the single step update comes from the natural Lagrange multiplier condition as shown in equation (16), equation (19) guarantees the resulted invariant algorithm remains the same. A simple solution can be obtained with scalars $\lambda_0, Q_1^t \in \mathbb{R}$ and vectors $s_m^t, s_c^t, Q_2^t, Q_3^t \in \mathbb{R}^n$. We use $\circ$ to denote $v_1 \circ v_2 \equiv v_1 v_2^T + v_2 v_1^T$ for two vectors $v_1, v_2 \in \mathbb{R}^n$.

$$\tilde{\nabla}_m M^t(\theta) = \lambda_0(s_m^t + m^t - m) \tag{20}$$

$$\tilde{\nabla}_c M^t(\theta) = \lambda_0((s_c^t + m^t - m)(s_c^t + m^t - m)^T - C) \tag{21}$$
$$+ Q_1^t + Q_2^t \circ m + Q_3^t m m^T$$

Corresponding updates for hyperparameter $\lambda_0 \in \mathbb{R}$ and self-evolved terms $s_m^t, s_c^t, Q_2^t, Q_3^t \in \mathbb{R}^n$ that initially zero are shown below. For simplicity needs, we denote $d_i^t \equiv x_i^t - m^t, d_w^t \equiv \sum_i \hat{w}_i d_i^t, \hat{d}_w^t \equiv d_w^t + m^t$ to represent statistics in a single generation, and $\hat{s}_m^{t-1} \equiv s_m^{t-1} + m^{t-1}, \hat{s}_c^{t-1} \equiv s_c^{t-1} + m^{t-1}$ to represent elements for historical information.

$$\lambda = \lambda_0/\lambda_0 + 1 \tag{22}$$

$$s_m^t + m^t = \lambda \hat{s}_m^{t-1} + (1 - \lambda)\hat{d}_w^{t-1} \tag{23}$$

$$s_c^t + m^t = \sqrt{\lambda}\hat{s}_c^{t-1} + \sqrt{1-\lambda}\hat{d}_w^{t-1} \tag{24}$$

$$Q_1^t = \lambda Q_1^{t-1} + \lambda \sum \hat{w}_i(d_i^{t-1} - d_w^{t-1})(d_i^{t-1} - d_w^{t-1})^T - \lambda_0\sqrt{\lambda}\sqrt{1-\lambda}\hat{d}_w^{t-1} \circ \hat{s}_c^{t-1} \tag{25}$$

$$Q_2^t = \lambda Q_2^{t-1} - \lambda_0(\sqrt{\lambda} + \sqrt{1-\lambda} - 2)(\sqrt{\lambda} * \hat{s}_c^{t-1} + \sqrt{1-\lambda} * \hat{d}_w^{t-1}) \tag{26}$$

$$Q_3^t = \lambda Q_3^{t-1} - \lambda_0(\sqrt{\lambda} - 1)(\sqrt{1-\lambda} - 1) \tag{27}$$

We now arrive at the single step update for next parameter $\theta^{t+1} = (m^{t+1}, C^{t+1})$. The resulting algorithm is named as SynCMA to emphasize another prominent characterization, the synchronous update nature, as discussed in section 3.2, besides invariance. The final updates in single iteration with $z_m = \eta_m + \lambda_0 + 1, z_c = \eta_c + \lambda_0 + 1, \beta^t = \frac{1}{z_m}(d_w^t + \lambda_0 s_m^t)$ for brevity sake is shown below.

$$m^{t+1} = m^t + \beta^t \tag{28}$$

$$C^{t+1} = \frac{\eta_c}{z_c}(C^t + \beta^t(\beta^t)^T) + \frac{\lambda_0}{z_c}(s_c^t - \beta^t)(s_c^t - \beta^t)^T \tag{29}$$
$$+ \frac{1}{z_c}(\sum_i \hat{w}_i(d_i^t - \beta^t)(d_i^t - \beta^t)^T + Q_1^t + Q_2^t \circ m^t + Q_3^t m^t(m^t)^T)$$

## 3.2 Theoretical Comparison with Other CMA Optimizers

**Theoretical advantages**. We here list four main theoretical advantages of SynCMA over other CMA optimizers : it is invariant, it has no external learning rate, it fully incorporates historical information, and it is synchronous.

The invariant, external learning rate free, and historical information fully incorporated properties are grant by induced framework INVIGO with historical information. For other CMA optimizers, they mostly stem from parameterization $\theta = (m, \sigma\Sigma)$ with $\sigma$ the external learning rate that need to be tuned or set rules Hansen (2016) for evolution. Also, in the previous literature, no optimizers ever incorporate historical information into the mean update with a stable optimization procedure. And our fully incorporation, actually decrease the number of parameters needed to tune in historical part.

The synchronous property from which SYNCMA is named is a result of the fact that INVIGO with historical information is built on IGO, which treats the current distribution $\theta$ as a single point in space to update, i.e. the updates for mean and covariance are intertwined. For other CMA algorithms, the updates are performed sequentially in each generation, e.g. $m^{t+1} = U_m(m^t, \sigma^t\Sigma^t), \Sigma^{t+1} = U_c(m^{t+1}, \sigma^t\Sigma^t)$, etc. Moreover, to strictly follow the proposition 1, updates need to be intertwined as SYNCMA does.

**Connection to CMA-ES**. It is also worth making the connection between SYNCMA and the CMA family algorithms. Where the approximations used correspond to the four theoretical advantages of SYNCMAmentioned above.

**Proposition 2.** *When (1) the historical information is partially used for covariance, i.e.* $\tilde{\nabla}_m M^t(\theta) = 0$ *and* $\tilde{\nabla}_c M^t(\theta) = \lambda_0((s_c^t + m^t - m)(s_c^t + m^t - m)^T - C)$. *(2) all the higher order terms, when assuming* $\eta_c \approx z_c \gg 1, z_m \gg 1$*, are discarded.* SYNCMA *coincide with CMA-ES up to an external learning rate difference.*

**Limitation and Future Direction**. There is still much to explore in both the INVIGO and the SYNCMA . For the framework, it is tempting to generalize to a broader family of models through such approximations. For the algorithm, we currently set parameters $\eta_m, \eta_c$ and $\lambda_0$ constants as in below experiments, which might make SYNCMA overshoot in the final stage of optimization when it is close to the optimum. However, they can absolutely be invariant variables instead of constants for a better overall optimization performance.

## 4 EXPERIMENTS

In this section, we evaluate SYNCMA with other baselines in synthetic functions, Mujoco locomotion tasks, and rover planning task. The criteria are chosen in the context of online optimization, focusing on full optimization procedures in the natural axis and near global optimal efficiency. We have therefore plotted all optimization procedures with the shaded area bounded by quantiles and the solid line denoting the median performance over all trails.

Baselines are chosen in a structured way. First, random search (RS) (Bergstra & Bengio, 2012) is chosen as the overall baseline. Then, two black-box optimizers, differential evolution (DE) (Storn & Price, 1997) and simulated annealing (SA) (Bouttier & Gavra, 2019) are chosen. Among the CMA optimizers, we choose CMA-ES and two of its leading variants DD-CMA and TR-CMA-ES for detailed comparison. Finally, the Bayesian optimization method TuRBO (Eriksson et al., 2019) is used as the state-of-the-art baseline for BO. Parameters $\eta_m, \eta_c$ of SYNCMAare set to constant that match the initial settings of the corresponding parameters in CMA-ES. $\lambda_0$ corresponds to a combination of several parameters in CMA-ES so we simply test with the constant value $\lambda_0 = 2$, which corresponds to the approximate counterpart in CMA-ES. We use this value throughout the main paper while there are better performances of SYNCMA with different $\lambda_0$ as shown in ablation studies in Appendix B.4.

TR-CMA-ES is based on its original paper version implemented in Matlab due to precision problems in Python, and therefore we exclude TR-CMA-ES in the Mujoco locomotion and rover planning tasks as they are based on specific Python libraries. All other baselines are implemented with their fine-tuned version available online (Duan et al., 2022; Balandat et al., 2020). See Appendix B for details.

### 4.1 SYNTHETIC FUNCTION

We select 10 commonly used synthetic functions with dimension $n$ arbitrarily set. These functions, including different characteristics such as multi-model, ill-conditioned and ill-scaled, are scaled to a global minimum value 0 with shifted domain. The batch size is $N = 2n$ and the evaluation limit

is the same for all optimizers except TuRBO, where the budget is fixed at 5,000 evaluations due to memory limitations.

The full experiments are run with different dimensions of $n = \{32, 64, 128\}$ and the results are presented in two ways under the same evaluation budget: near global optimum performance and the whole optimization procedure. Limited results are presented here and please refer to Appendix B.1 for the full contents.

Table 1: Near global optimum performance on 64d synthetic functions(lower is better) over 20 trials with budget of 50000 evaluations, TuRBO is excluded. Numbers in brackets indicates the median evaluation number needed for optimizers to achieve value better than 0.5.

| Optimizer | Sphere | Discus | Schwefel | DiffPowers | LevyMontalvo | Rastrigin | Ackley |
|---|---|---|---|---|---|---|---|
| SA | 0.6 | 100.7 | 0.1(2650) | 37.5 | 6.8 | 1634.7 | 13.6 |
| RS | 793.9 | 1138.8 | 29902.6 | 2369.4 | 14.4 | 1395.9 | 11.8 |
| DE | 7.2 | 20.0 | 25.6 | 63.6 | 0.4(49700) | 586.9 | 3.1 |
| DDCMA | 0.0(10047) | **0.0(12748)** | 0.0(16820) | 0.0(10164) | 0.0(9087) | 17.9 | 0.0(11757) |
| CMAES | 0.0(13825) | 0.0(43265) | 0.0(16134) | 0.0(18503) | 0.0(9901) | 21.4 | 0.0(15620) |
| TRCMAES | 0.0(7185) | 85.9 | 0.0(9905) | 0.0(12040) | 0.0(5515) | 22.4 | 0.0(7869) |
| SYNCMA(**Ours**) | **0.0(3938)** | 0.0(18820) | **0.0(1157)** | **0.0(1158)** | **0.0(2318)** | **0.2(42696)** | **0.0(7567)** |

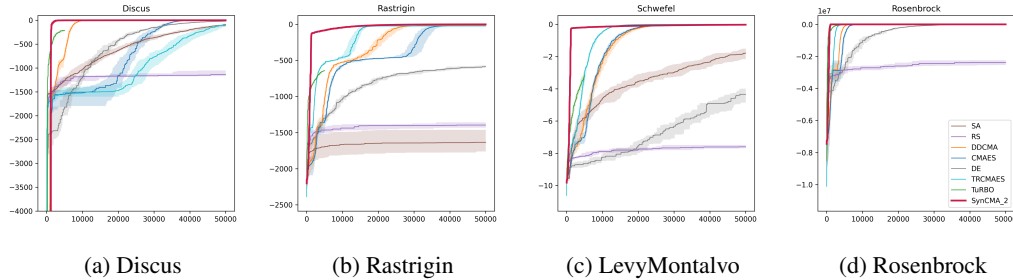

| (a) Discus | (b) Rastrigin | (c) LevyMontalvo | (d) Rosenbrock |
|---|---|---|---|

Figure 1: Optimization procedure in 4 typical synthetic functions with dimension $n = 64$ over 20 trails considering all optimizers. Index of SYNCMA indicate $\lambda_0$.

According to table 1 where TuRBO is excluded as it is unable to scale to this budget, SYNCMA demonstrate both superior optimization capability and efficiency over others. While other optimizers are less efficient and fail to optimize high-conditional number multi-model function Rastrigin, ill-scaled function Discus and others. Further, full optimization procedures including TuRBO with maximum budget under storage limit are partially shown in figure 1, SYNCMA still outperforms others including TuRBO after first several hundreds evaluation from 32 to 128 dimension, demonstrating the capability of such optimizer derived from an invariant framework.

## 4.2 MUJOCO LOCOMOTION TASK

After the synthetic functions, we evaluate SYNCMA and other baselines on the more realistic Mujoco locomotion tasks (Todorov et al., 2012), which are popular benchmarks for reinforcement learning algorithms. To run sampling-based optimizers on Mujoco, we refer to (Mania et al., 2018) and optimize a linear policy: $\mathbf{a} = \mathbf{W}\mathbf{s}$, where $\mathbf{a}$ is the agent action and $\mathbf{s}$ is the environment state. The parameter matrix $\mathbf{W}$ are continuous and in the range of $[-1, 1]$. Among all 6 tasks, we dismiss the overly high dimensional task Humanoid(6392d) and test all other 5 tasks with batch size $N = 100$. Two results are shown here in figure 2a, 2b with more results in Appendix B.2. While TuRBO dominates other baselines, SYNCMA outperforms TuRBO in 2 tasks and remains competitive with TuRBO for the other 3 tasks.

## 4.3 ROVER PLANNING TASK

To further explore the empirical performance of SYNCMA in a realistic setting, we consider the rover trajectory optimization task, where a start position $s$ and a goal position $g$ are defined in the

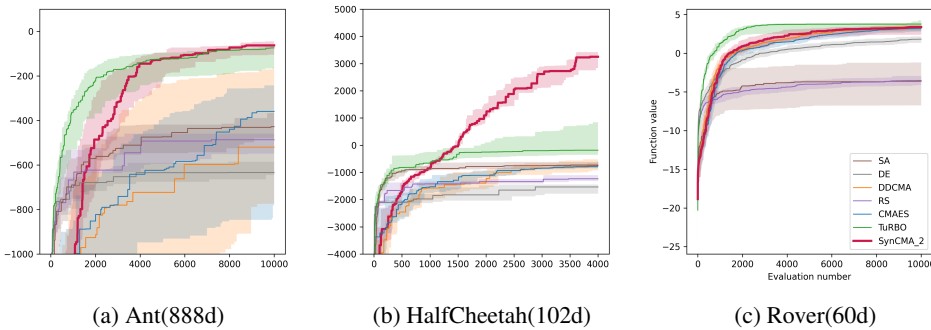

(a) Ant(888d)        (b) HalfCheetah(102d)        (c) Rover(60d)

Figure 2: Optimization procedure for two high dimensional Mujoco locomotion tasks over 10 trials and rover planning task over 100 trails. Index of SYNCMA indicate $\lambda_0$.

2D plane, as well as a cost function $c(x)$ over the state space. The trajectories are described by a set of points to which a B-spline is fitted and the cost function is computed. The whole state space is $x \in [0, 1]^{60}$ and we make the batch size $N = 2n = 120$. A reward function to be optimized is defined to be non-smooth, discontinuous, and concave over the first two and last two dimensions of the state. The result in figure 6f shows that SYNCMA still exhibits competitive performance over other baselines.

## 4.4 ABLATION STUDY

The weight for historical information $\lambda_0$ is a parameter that substitutes a combination of several parameters in CMA optimizers, and is set constantly as $\lambda_0 = 2$. We thus study the sensitivity on this parameter for constant setting here. All of previous experiments are repeated for $\lambda_0 \in [0, 4]$, with results for $\lambda_0 = \{0, 1, 2, 4\}$ shown in Appendix B.4, from which we summarize several observations within range $[0, 4]$ here.

**Sensitivity**. When SYNCMA includes historical information, i.e. $\lambda_0 > 0$, SYNCMA consistently shows competitive performance.

**Function Landscape**. When their exists a fundamental subspace that covers the structure of the problem, as in Rastrigin, a higher $\lambda_0$ yields better performance and efficiency. Otherwise, as in LevyMontalvo, a higher $\lambda_0$ might be detrimental.

**Dimensionality**. Observed from tasks in Mujoco, synthetic functions, and rover planning, a higher dimension generally requires a higher $\lambda_0$.

## 5 CONCLUSION

We present an invariant optimizer framework INVIGO with no external learning rate and accessibility to fully incorporate historical information. Although the framework is built on the assumption and computational cost of information geometric optimization, we expect the possible generalization to a wider family of models because of the approximations used in the formulation and solving stages. When exemplified with multi-dimensional Gaussian, our framework derives a competitive scalable optimizer SYNCMA for both synthetic and realistic scenarios, even when the parameters for updates are constant.

The outperformance of SYNCMA over leading Bayesian optimizers can mean a lot to nature-inspired optimizers and the field of local search. Its competence not only demonstrates the power of the invariant property derived from a complete framework, but also suggests the under-explored potential of local search over global search methods such as Bayesian optimization. The intuition for the latter assertion is that if the seemingly high dimension problem has a lower subspace that really matters, then local search may find it more efficiently. If the problem is fundamentally in high dimension, then global search methods are naturally limited due to computational constraints.

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

APPENDICES : AN INVARIANT INFORMATION GEOMETRIC METHOD FOR HIGH-DIMENSIONAL ONLINE OPTIMIZATION

# A    PROOFS AND DERIVATIONS

## A.1    PROOF FOR THEOREM 1

*Proof.* To prove the former part, it is suffice to notice,

$$\nabla_\theta \log L_{\theta^t}(\theta) - (-\nabla_\theta D_{KL}(q_{\theta^t}||p_\theta)) = \nabla_\theta D_{KL}(q_{\theta^t}||q_\theta) \tag{30}$$

$$= \nabla_\theta(\frac{1}{2}\sum I_{ij}^{q_{\theta^t}}(\theta^t)\delta\theta_i\delta\theta_j + O(\delta\theta^3)) \tag{31}$$

To prove the latter part, we first apply natural gradient to $L_{\theta^t}(\theta)$,

$$\tilde{\nabla}_\theta L_{\theta^t}(\theta) = \tilde{\nabla}_\theta \int g_{f,\theta^t}(x)p_\theta(x)\mathrm{d}x \tag{32}$$

$$= I^{-1}(\theta)\int g_{f,\theta^t}(x)\frac{\partial \ln p_\theta(x)}{\partial\theta}\frac{p_\theta(x)}{p_{\theta^t}(x)}p_{\theta^t}(\mathrm{d}x) \tag{33}$$

In contrast, applying natural gradient to $D_{KL}(q_{\theta^t}||p_\theta)$,

$$\tilde{\nabla}_\theta D_{KL}(q_{\theta^t}||p_\theta) = -I^{-1}(\theta)\int \frac{g_{f,\theta^t}(x)}{E_{p_{\theta^t}}[g_{f,\theta^t}(x)]}\frac{\partial \ln p_\theta(x)}{\partial\theta}p_{\theta^t}(\mathrm{d}x) \tag{34}$$

Thus under assumption 1 , only $D_{KL}(q_{\theta^t}||p_\theta)$ is available to take natural gradient on any point while the computational cost remain the same ☐

## A.2    PROOF FOR THEOREM 2, 3

*Proof.* It is suffice to prove theorem 3 as InvIGO with historical information recovers to In-vIGO when $\lambda$ is zero. Similar to $\theta$, we define $q'_{\varphi^t}(x) = \frac{p'_{\varphi^t}(x)g_{f,\varphi^t}(x)}{\mathbb{E}_{p'_{\varphi^t}}[g_{f,\varphi^t}(x)]}$.

Initially, we have $q'_{\varphi^0}(x) = q_{\theta^0}(x)$ from given conditions. So we assume for the current time $t$, $q'_{\varphi^t}(x) = q_{\theta^t}(x)$ holds as well. Then from equation (16), we have

$$\tilde{\nabla}_\varphi|_{\varphi=\varphi^{t+1}}(-\sum_{\tau=0}^{T}\lambda^\tau D_{KL}(q'_{\varphi^{t-\tau}}||p'_\varphi) + \eta(\epsilon^2/2 - D_{KL}(p'_{\varphi^t}||p'_\varphi))) \tag{35}$$

$$= \tilde{\nabla}_\theta|_{\theta=\varphi^{-1}(\varphi^{t+1})}(-\sum_{\tau=0}^{T}\lambda^\tau D_{KL}(q_{\theta^{t-\tau}}||p_\theta) + \eta(\epsilon^2/2 - D_{KL}(p_{\theta^t}||p_\theta))) \cdot (\frac{\partial\theta}{\partial\varphi}|_{\theta=\varphi^{-1}(\varphi^{t+1})})^3 \tag{36}$$

$$= 0 \tag{37}$$

Thus we have $\varphi^{t+1} = \varphi(\theta^{t+1})$. From mathematical infuction, our proof finished. ☐

## A.3    DERIVATIONS OF SYNCMA

The derivations consist of two parts. The first part is to substitute optimization objevtive $G^t(\theta) = \sum_{\tau=0}^{t-1}\lambda^\tau D_{KL}(q_{\theta^{t-\tau}}||p_\theta)$ with a self-evolved term $M^t(\theta)$ so that the gradient information is completely preserved as equation (19) while the computational costs reduce to the same as IGO, i.e. the cost of computing $\tilde{\nabla}_\theta|_{\theta=\theta^t}L_{\theta^t}(\theta)$. The second part is to derive analytical updates for $\theta^{t+1} = (m^{t+1}, C^{t+1})$ from solving equation (16).

### A.3.1 Substitution of $G^t(\theta)$

We start from our target of preserving historical information,
$$\tilde{\nabla}_\theta G^t(\theta) = -\tilde{\nabla}_\theta M^t(\theta) + \tilde{\nabla}_\theta D_{KL}(q_{\theta^t}||p_\theta) \tag{38}$$

We assume $M^t(\theta)$ has the form that present in the main paper, with scalars $\lambda_0, Q_1^t \in \mathbb{R}$ and vectors $s_m^t, s_c^t, Q_2^t, Q_3^t \in \mathbb{R}^n$ all start from zero. We use $\circ$ to denote $v_1 \circ v_2 \equiv v_1 v_2^T + v_2 v_1^T$ for two vectors $v_1, v_2 \in \mathbb{R}^n$.
$$\tilde{\nabla}_m M^t(\theta) = \lambda_0(s_m^t + m^t - m) \tag{39}$$
$$\tilde{\nabla}_c M^t(\theta) = \lambda_0((s_c^t + m^t - m)(s_c^t + m^t - m)^T - C) \tag{40}$$
$$+ Q_1^t + Q_2^t \circ m + Q_3^t mm^T$$

Notice that $G^t(\theta) = \lambda G^{t-1} + D_{KL}(q_{\theta^t}||p_\theta)$, therefore we have,
$$\tilde{\nabla}_\theta M^t(\theta) = \lambda(\tilde{\nabla}_\theta M^{t-1}(\theta) - \tilde{\nabla}_\theta D_{KL}(q_{\theta^{t-1}}||p_\theta)) \tag{41}$$

**Mean Solution**   . Apply assumption (39) for $\tilde{\nabla}_m M^t(\theta)$ yields,
$$\lambda_0(s_m^t + m^t - m) = \lambda \lambda_0(s_m^{t-1} + m^{t-1} - m) + \lambda \sum_i \hat{w}_i(x_i^{t-1} - m) \tag{42}$$

Which straightly gives the only solutions on $\lambda$ and $s_m^t$,
$$\lambda = \frac{\lambda_0}{\lambda_0 + 1} \tag{43}$$
$$s_m^t + m^t = \lambda s_m^{t-1} + (1 - \lambda)d_w^{t-1} + m^{t-1} \tag{44}$$

**Covariance Maxtrix Solution**   Apply assumption (40) for $\tilde{\nabla}_c M^t(\theta)$ while denoting $Q^t(m) = Q_1^t + Q_2^t \circ m + Q_3^t mm^T$ yields,
$$\lambda_0((s_c^t + m^t - m)(s_c^t + m^t - m)^T - C) + Q^t(m) \tag{45}$$
$$= \lambda \lambda_0((s_c^{t-1} + m^{t-1} - m)(s_c^{t-1} + m^{t-1} - m)^T - C) + \lambda Q^{t-1}(m)$$
$$+ \lambda(\sum_i \hat{w}_i(x_i^{t-1} - m)(x_i^{t-1} - m)^T - C)$$

To cancel out $C$ on both side, same value of $lambda = \lambda_0/\lambda_0+1$ is derived. Further, we assume the form of update on $s_c^t$ is similar to $s_m^t$ with parameter $\alpha$ and $zeta$ undetermined yet,
$$s_c^t + m^t = \zeta(s_c^{t-1} + m^{t-1}) + \alpha(d_w^{t-1} + m^{t-1}) \tag{46}$$

Thus we arrive at,
$$(s_c^t + m^t - m)(s_c^t + m^t - m)^T = \zeta^2(s_c^{t-1} + m^{t-1} - m)(s_c^{t-1} + m^{t-1} - m)^T \tag{47}$$
$$+ \alpha^2(d_w^{t-1} + m^{t-1} - m)(d_w^{t-1} + m^{t-1} - m)^T$$
$$+ (\alpha + \zeta - 2)m \circ (\zeta(s_c^{t-1} + m^{t-1}) + \alpha(d_w^{t-1} + m^{t-1}))$$
$$+ (\alpha - 1)(\zeta - 1)mm^T$$
$$+ \alpha\zeta(d_w^{t-1} + m^{t-1}) \circ (s_c^{t-1} + m^{t-1})$$

Notice that for the second term of Eq.47, we have,
$$(d_w^{t-1} + m^{t-1} - m)(d_w^{t-1} + m^{t-1} - m)^T$$
$$= \mathbb{E}_{q_{\theta^{t-1}}}[x - m]\mathbb{E}_{q_{\theta^{t-1}}}[x - m]^T \tag{48}$$
$$= \mathbb{E}_{q_{\theta^{t-1}}}[(x - \mathbb{E}_{q_{\theta^{t-1}}}[x] + \mathbb{E}_{q_{\theta^{t-1}}}[x] - m)(x - \mathbb{E}_{q_{\theta^{t-1}}}[x] + \mathbb{E}_{q_{\theta^{t-1}}}[x] - m)^T] \tag{49}$$
$$- \mathbb{E}_{q_{\theta^{t-1}}}[(x - \mathbb{E}_{q_{\theta^{t-1}}}[x])(x - \mathbb{E}_{q_{\theta^{t-1}}}[x])^T]$$
$$= \mathbb{E}_{q_{\theta^{t-1}}}[(x - m)(x - m)^T] - \mathbb{E}_{q_{\theta^{t-1}}}[(x - \mathbb{E}_{q_{\theta^{t-1}}}[x])(x - \mathbb{E}_{q_{\theta^{t-1}}}[x])^T] \tag{50}$$
$$= \sum_i \hat{w}_i(x_i^{t-1} - m)(x_i^{t-1} - m)^T - \sum_i \hat{w}_i(d_i^{t-1} - d_w^{t-1})(d_i^{t-1} - d_w^{t-1})^T \tag{51}$$

Thus, to meet Eq.45, we arrive at updates for $s_c^t, Q_1^t, Q_2^t, Q_3^t$ with $\zeta = \sqrt{\lambda}, \alpha = \sqrt{1-\lambda}$,

$$s_c^t + m^t = \zeta s_c^{t-1} + \alpha d_w^{t-1} + (\zeta + \alpha)m_{t-1} \tag{52}$$

$$Q_1^t = \lambda Q_1^{t-1} + \lambda \sum \hat{w}_i (d_i^{t-1} - d_w^{t-1})(d_i^{t-1} - d_w^{t-1})^T \tag{53}$$
$$- \lambda_0 \alpha \zeta (d_w^{t-1} + m^{t-1}) \circ (s_c^{t-1} + m^{t-1})$$

$$Q_2^t = \lambda Q_2^{t-1} - \lambda_0 (\zeta + \alpha - 2)(\zeta * (s_c^{t-1} + m^{t-1}) + \alpha * (d_w^{t-1} + m^{t-1})) \tag{54}$$

$$Q_3^t = \lambda Q_3^{t-1} - \lambda_0 (\zeta - 1)(\alpha - 1) \tag{55}$$

### A.3.2 DERIVATION OF $\theta^{t+1}$

Substitute the $G^t(\theta)$ in the natural Lagrange condition (16) gives equation,

$$0 = \tilde{\nabla}_\theta|_{\theta=\theta^{t+1}}(-G^t(\theta) + \eta(\epsilon^2/2 - D_{KL}(p_{\theta^t}||p_\theta))) \tag{56}$$
$$= \tilde{\nabla}_\theta|_{\theta=\theta^{t+1}}(M^t(\theta) - D_{KL}(q_{\theta^t}||p_\theta) - [\eta_m, \eta_c]^T D_{KL}(p_{\theta^t}||p_\theta)) \tag{57}$$

Apply proposition 1 on above equation, we now arrive at equations for $\theta^{t+1}$,

$$\sum_i (\hat{w}_i^t + \frac{\eta_m}{N})(x_i^t - m^{t+1}) + \tilde{\nabla}_m M^t(\theta) = 0 \tag{58}$$

$$\sum_i (\hat{w}_i^t + \frac{\eta_c}{N})((x_i^t - m^{t+1})(x_i^t - m^{t+1})^T - C^{t+1}) + \tilde{\nabla}_c M^t(\theta) = 0 \tag{59}$$

Solving equations above straightly give updates with $z_m = \eta_m + \lambda_0 + 1, z_c = \eta_c + \lambda_0 + 1, \beta^t = \frac{1}{z_m}(d_w^t + \lambda_0 s_m^t)$ for brevity sake.

$$m^{t+1} = m^t + \beta^t \tag{60}$$

$$C^{t+1} = \frac{\eta_c}{z_c}(C^t + \beta^t(\beta^t)^T) + \frac{\lambda_0}{z_c}(s_c^t - \beta^t)(s_c^t - \beta^t)^T \tag{61}$$
$$+ \frac{1}{z_c}(\sum_i \hat{w}_i(d_i^t - \beta^t)(d_i^t - \beta^t)^T + Q_1^t + Q_2^t \circ m^t + Q_3^t m^t (m^t)^T)$$

### A.4 PROOF AND DISCUSSIONS ON PROPOSITION 2

*Proof.* We apply the first condition to equations (58, 59), which gives the updates for $\theta^{t+1}$ with $D_w^t = \sum_i \hat{w}_i^t (x_i^t - m^t)(x_i^t - m^t)^T$,

$$m^{t+1} = m^t + \frac{1}{z_m} d_w^t \tag{62}$$

$$C^{t+1} = \frac{\eta_c}{z_c} C^t + \frac{1}{z_c} D_w^t + \frac{\lambda_0}{z_c} s_c^t (s_c^t)^T \tag{63}$$
$$- \frac{\lambda_0}{z_c z_m}(d_w^t \circ (s_c^t)) - \frac{1}{z_c z_m}(2 - \frac{z_c}{z_m}) d_w^t (d_w^t)^T$$

Then under the second condition, the last two terms in the update of $C^{t+1}$ should be discarded. The rest part coincide with CMA-ES up to an external learning rate difference. $\square$

## B EXPERIMENTS

We implement SYNCMA using PyPop7 (Duan et al., 2022)[1]. All baselines except TuRBO and TR-CMA-ES are also implemented with this library. Except for the hyperparameters mentioned in the main paper, all evolutionary based algorithms use default options. In SYNCMA, we use constant learning rate $\sigma = 0.1$ in order to match the initial learning rate in other CMA optimizers to sample

---

[1] https://github.com/Evolutionary-Intelligence/pypop

with $\mathcal{N}(m, \sigma\Sigma)$. As CMA optimizers also use $\sigma$ as the update fraction for mean, we set $\eta_m$ such that $1/z_m = 0.1$ accordingly. $\eta_c$ is set two times of the corresponding value in CMA-ES, as several tunning techniques used in fine-tuned version of CMA-ES may let its actual corresponding value larger. For TuRBO, we refer to the implementation of BoTorch and replicate the original algorithm. We set the trust region number as one and keep all hyperparameter same as the original implementation.[2] All experiments are held on Intel(R) Xeon(R) Platinum 8180 CPU @ 2.50GHz, except for TuRBO method, a NVIDIA A100 is used.

## B.1 SYNTHETIC FUNCTIONS

For synthetic functions, we directly use the benchmarks provided by PyPop7, with formulation shown in Table 2. Full contents of near optimal performance and the average efficiency rank, i.e. the first time hitting time into 0.5 value, for CMA optimizers under the same budget are provided in tables 3, 4 and 5, other optimizers are excluded as their poor performance and linewidth limit. The optimization procedure of all 10 functions for SYNCMA with different $\lambda_0$ and other baselines are provided in figures 3, 4, 5.

It can be seen that as dimension increasing, the average performance of SYNCMA is getting worse. This align with the observation on dimensionality in ablation study that higher dimension generally yields higher $\lambda_0$.

Table 2: 10 different synthetic functions. Selected from classic test function for global optimization, including different rugged characteristics like multi-modality, high condition number and different optima landscape. The optima all scale to $x^* = 0$, $f(x^*) = 0$. For the brevity sake, $w_i = 1 + \frac{x_i+1}{4}$ in LevyMontalvo function, $z_i = x_i^2 + x_{i+1}^2$ in Schaffer function.

| Name | Expression |
|------|------------|
| Sphere | $\sum_i x_i^2$ |
| Discus | $10^6 x_0 + \sum_{i \geq 1} x_i$ |
| Schwefel | $\sum_i x_i + \prod_i x_i$ |
| DiffPowers | $\sum_i x^{2+4i/n}$ |
| Bohachevsky | $\sum_{i<n-1} 0.7 + x_i^2 + 2*x_{i+1}^2 - 0.3\cos(3\pi x_i) - 0.4\cos(4\pi x_{i+1})$ |
| LevyMontalvo | $\frac{\pi}{n}(10\sin^2(\pi w_0) + (w_{n-1}-1)^2 + \sum_{i<n-1}(w_i-1)^2(1+10\sin^2(\pi w_{i+1})))$ |
| Rastrigin | $10n + \sum_i x_i^2 - 10\cos(2\pi x_i)$ |
| Ackley | $-20\exp(-0.2(\frac{1}{n}\sum_i x_i^2)^{0.5}) - \exp(\frac{1}{n}\sum_i \cos(2\pi x_i)) + 20 + e$ |
| Schaffer | $\sum_{i<n-1} z_i^{0.25} * (\sin^2(50*z_i^{0.1}) + 1)$ |
| Rosenbrock | $100\sum_{i \geq 1}(x_i - x_{i-1}^2)^2 + \sum_{i<n-1}(x_i-1)^2$ |

## B.2 MUJOCO LOCOMOTION TASK

For Mujoco locomotion benchmarks, we refer to ARS[3] to model the task as a sampling problem, rollout is set to 1 for simplicity. Full optimization procedure for all 5 tasks are provided in 6.

## B.3 ROVER PLANNING TASK

For rover planning task, we refer to nevergrad[4] to test. Optimization procedure is provided in 6f.

## B.4 ABLATION STUDY

All the data provided in appendix actually contains different version of SYNCMA with $\lambda_0 = \{0, 1, 2, 4\}$, so they naturally consist the ablation study. The three observations can be verified

---

[2] https://botorch.org/tutorials/turbo_1
[3] https://github.com/modestyachts/ARS
[4] https://github.com/facebookresearch/nevergrad

thereby. Through every experiment on each $\lambda_0$, we monitor the behavior of SYNCMA and do not find any degreneration happen.

Table 3: Near optimal performance of CMA optimizers on 32d synthetic functions(lower is better). We show the median best performance of 10,000 evaluations over 100 trials. Numbers in brackets indicates the evaluation number need for algorithms to achieve better than 0.5.

| Function | DD-CMA | CMA-ES | TR-CMA-ES | SynCMA_1 | SynCMA_2 | SynCMA_4 |
|---|---|---|---|---|---|---|
| Sphere | 0.0(3665) | 0.0(4612) | 0.1(2405) | 0.0(855) | 0.0(555) | 0.0(455) |
| Discus | 0.0(5223) | 52.8 | 419.6 | 0.1(6363) | 0.0(4437) | 0.0(2568) |
| Schwefel | 0.0(5325) | 0.0(5124) | 0.1(3074) | 0.0(704) | 0.0(519) | 0.0(449) |
| DiffPowers | 0.0(3703) | 0.0(5585) | 0.1(3720) | 0.0(704) | 0.0(519) | 0.0(450) |
| Bohachevsky | 0.0(5420) | 0.0(7115) | 0.4(4428) | 0.2(8163) | 0.1(6340) | 0.0(5106) |
| LevyMontalvo | 0.0(5153) | 0.0(4030) | 0.1(2506) | 0.0(1350) | 0.0(1092) | 0.1(1094) |
| Rastrigin | 58.1 | 194.8 | 15.9 | 1.0 | 0.2(8758) | 0.1(7372) |
| Ackley | 0.0(4849) | 0.0(5850) | 0.1(2946) | 0.1(2733) | 0.0(2133) | 0.0(1134) |
| Schaffer | 61.3 | 71.3 | 2.2 | 6.6 | 4.8 | 3.8 |
| Rosenbrock | 28.0 | 29.9 | 0.1(7251) | 29.3 | 29.2 | 29.6 |
| **Ave_Rank** | 5 | 6 | 4 | 3 | 2 | 1 |

Table 4: Near optimal performance of CMA optimizers on 64d synthetic functions(lower is better). We show the median best performance of 50,000 evaluations over 20 trials. Numbers in brackets indicates the evaluation number need for algorithms to achieve better than 0.5.

| Function | DD-CMA | CMA-ES | TR-CMA-ES | SynCMA_1 | SynCMA_2 | SynCMA_4 |
|---|---|---|---|---|---|---|
| Sphere | 0.0(10047) | 0.0(13825) | 0.0(7185) | 0.0(4691) | 0.0(3938) | 0.0(1047) |
| Discus | 0.0(12748) | 0.0(43265) | 85.9 | 0.0(27662) | 0.0(18820) | 0.0(10484) |
| Schwefel | 0.0(16820) | 0.0(16134) | 0.0(9905) | 0.0(1548) | 0.0(1157) | 0.0(929) |
| DiffPowers | 0.0(10164) | 0.0(18503) | 0.0(12040) | 0.0(1548) | 0.0(1158) | 0.0(937) |
| Bohachevsky | 0.0(14605) | 0.0(20369) | 0.0(11865) | 0.3(44088) | 0.0(32247) | 0.0(23314) |
| LevyMontalvo | 0.0(9087) | 0.0(9901) | 0.0(5515) | 0.0(2522) | 0.0(2318) | 0.0(2486) |
| Rastrigin | 17.9 | 21.4 | 22.4 | 1.3 | 0.2(42696) | 0.0(31565) |
| Ackley | 0.0(11757) | 0.0(15620) | 0.0(7869) | 0.0(9800) | 0.0(7567) | 0.0(4549) |
| Schaffer | 23.3 | 0.4(49400) | 0.3(32701) | 12.3 | 8.2 | 4.9 |
| Rosenbrock | 53.8 | 57.8 | 0.0(21543) | 59.8 | 60.1 | 60.9 |
| **Ave_Rank** | 5 | 6 | 3 | 4 | 2 | 1 |

Table 5: Near optimal performance of CMA optimizers on 128d synthetic functions(lower is better). We show the median best performance of 100,000 evaluations over 20 trials. Numbers in brackets indicates the evaluation number need for algorithms to achieve better than 0.5.

| Function | DD-CMA | CMA-ES | TR-CMA-ES | SynCMA_1 | SynCMA_2 | SynCMA_4 |
|---|---|---|---|---|---|---|
| Sphere | 0.0(39745) | 0.0(27751) | 0.0(21428) | 0.2(54617) | 0.1(40403) | 0.0(25428) |
| Discus | 395.8 | 0.0(32544) | 1496.4 | 1.0 | 0.4(93272) | 0.1(58425) |
| Schwefel | 0.0(51681) | 0.0(59947) | 0.0(31407) | 0.1(3610) | 0.1(2600) | 0.1(2085) |
| DiffPowers | 0.0(62675) | 0.0(28185) | 0.0(41019) | 0.0(3626) | 0.0(2621) | 0.0(2123) |
| Bohachevsky | 0.0(57138) | 0.0(39401) | 0.0(31801) | 9.8 | 4.8 | 1.2 |
| LevyMontalvo | 0.0(26924) | 0.0(20332) | 0.0(14875) | 0.0(5672) | 0.0(5404) | 0.0(5344) |
| Rastrigin | 107.6 | 23.0 | 33.3 | 42.6 | 19.9 | 5.0 |
| Ackley | 0.0(41488) | 0.0(29714) | 0.0(21994) | 0.3(46327) | 0.2(33839) | 0.1(20330) |
| Schaffer | 4.3 | 0.7 | 0.2(91494) | 45.1 | 37.7 | 27.9 |
| Rosenbrock | 124.3 | 120.5 | 0.0(68532) | 146.6 | 134.8 | 127.8 |
| **Ave_Rank** | 5 | 3 | 2 | 6 | 4 | 1 |

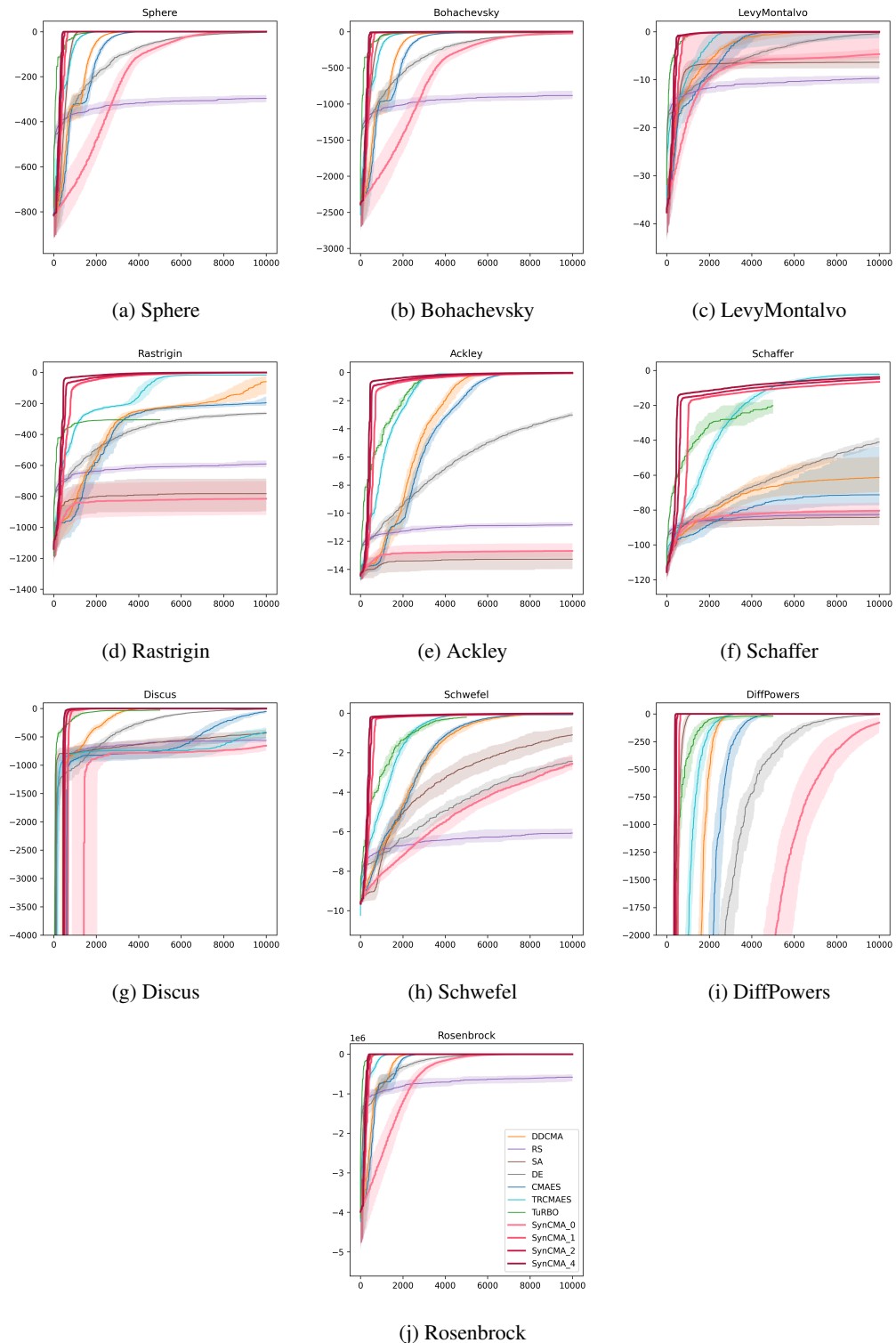

Figure 3: Optimization procedure in 10 tests function with dimension $n = 32$ over 100 trails with 10000 evaluations.

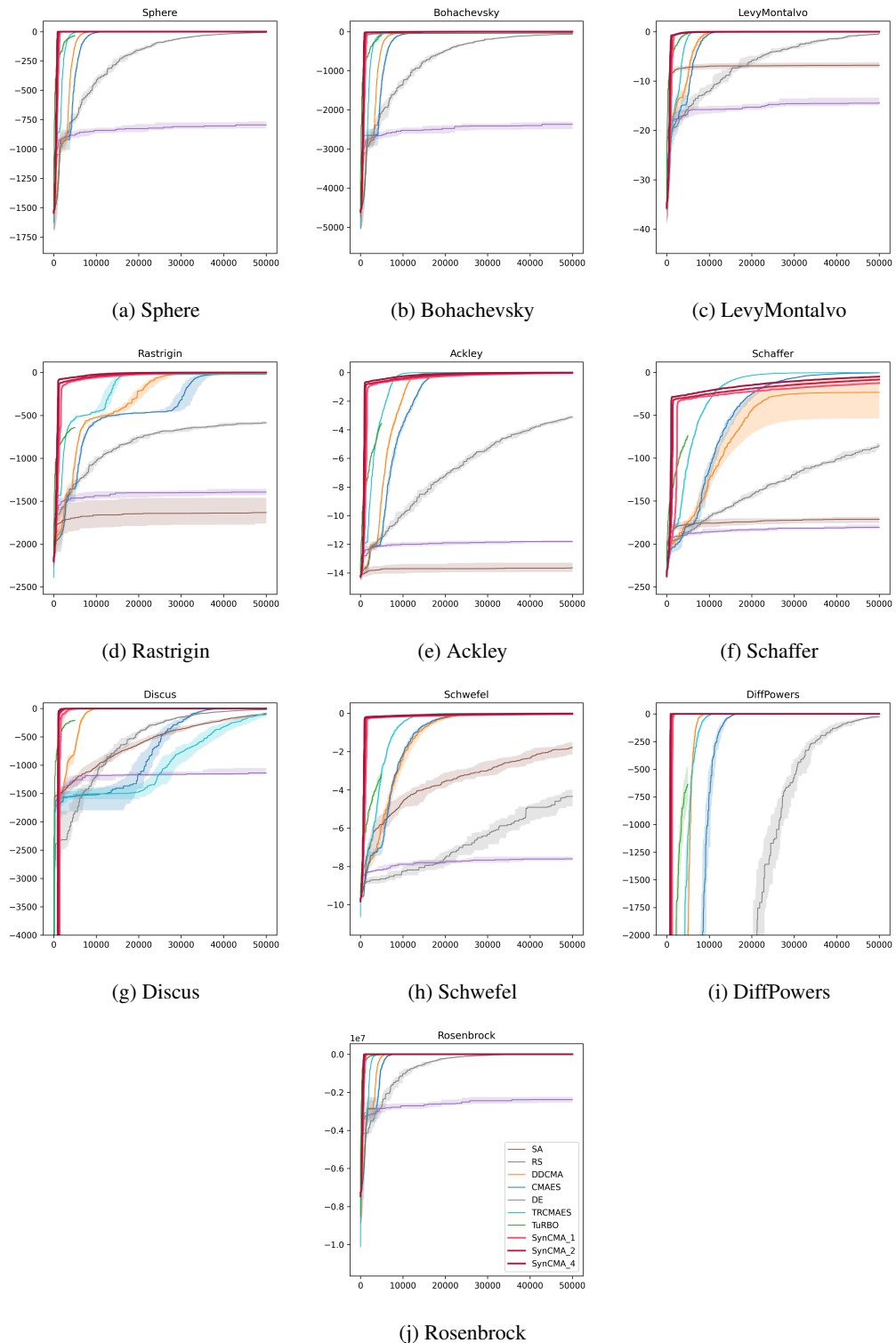

Figure 4: Optimization procedure in 10 tests function with dimension $n = 64$ over 100 trails with 50000 evaluations.

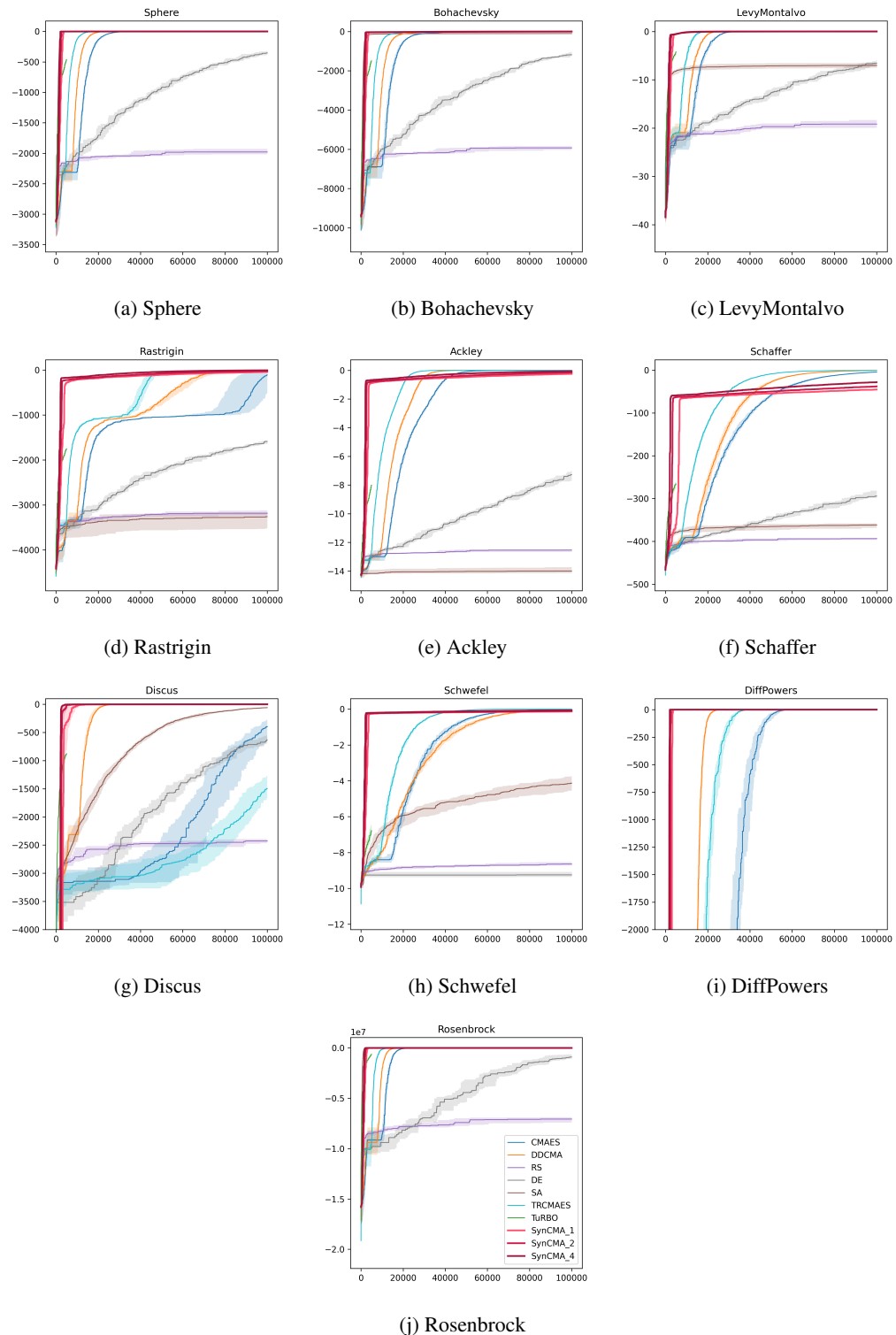

Figure 5: Optimization procedure in 10 tests function with dimension $n = 128$ over 100 trails with 100000 evaluations.

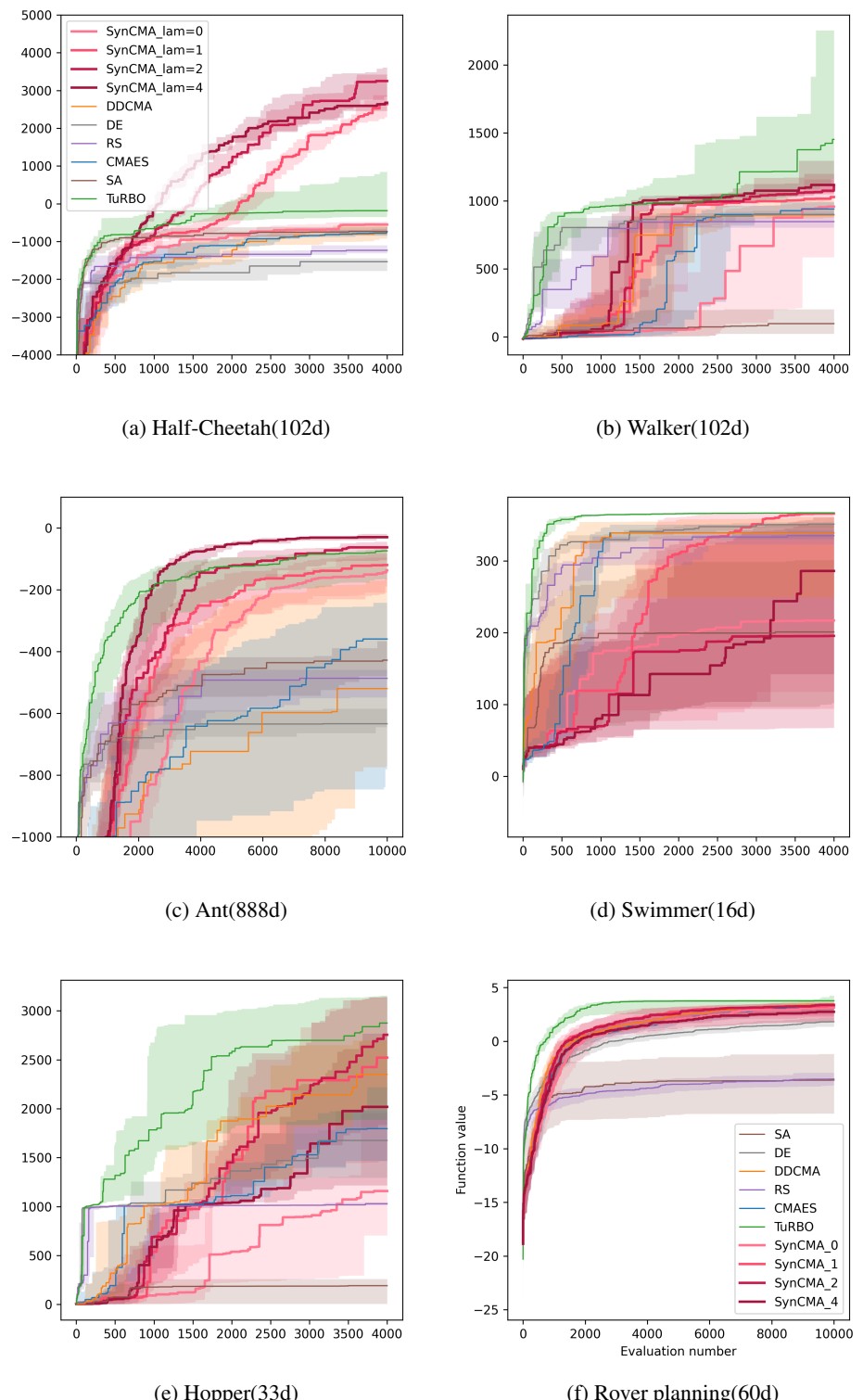

(a) Half-Cheetah(102d)

(b) Walker(102d)

(c) Ant(888d)

(d) Swimmer(16d)

(e) Hopper(33d)

(f) Rover planning(60d)

Figure 6: Optimization performance on 5 Mujoco locomotion tasks and the rover planing task

