# OpenReview forum: "An Invariant Information Geometric Method for High-dimensional Online Optimization"
_ICLR.cc/2024/Conference — ICLR 2024 Conference Withdrawn Submission_

### Official Review · Reviewer_wi31 · 2023-10-13

**Soundness:** 3 good
**Presentation:** 3 good
**Contribution:** 3 good
**Rating:** 5
**Confidence:** 1

**Summary:**

This paper propose a new online learning method based on evolutionary strategy. The proposed method is geometric invariant under transformations.

**Strengths:**

The whole paper is well-presented and easy to follow.

**Weaknesses:**

see questions.

**Questions:**

1. The theoretical guarantee of the proposed method seems weak. The theory should be formulated as Theorems based clear assumptions. Current statements in the theory section are vague.

2. Limitations and further directions should be included in the last section?

3. Theorem 1 looks like just some properties.

4. Theorem 2/3 looks very vague.

5. Why use such a long section (section 3) just as an example? Why only focusing studying the more general case?

6. Overall, though I am not familiar with the studied topic, I think combining geometric invariant with ES seems not very significant, since the fisher matrix already has such property.

7. Do use historic data a standard way in online learning? Does other method also use such technique?



minors:
problem 12 -> problem (12)

---

> ### Author Response · Authors · 2023-11-20
> **Reply to your concerns**
>
> Thank you for your detailed and insightful review, which provides a second perspective on our work. For a better discussion, we would like to address your concerns in turn.
>
> Responses to Questions :
>
> - (A1) We admit that the theoretical guarantees are weak compared with some existing work on online optimization , and are considering changing the theorems to propositions. However, comparing with other first-order method in the black-box setting with arbitrary objective which is too hard for a regret or convergence bound, our results match the theoretical strength. Also, we perceive our work as a clear theoretical framework plus a derived powerful algorithms, of which we are inclined to focus on the latter.
>
> - (A2) We'll definitely consider your suggestion! Putting the limitation and future directions in the last section will give more room for an overall discussion. Currently, it is in Section 3.3 to follow with theoretical comparisons of SynCMA. As the same reason mentioned in (A1), in black-box objective, people usually tend to focus on a practical method with comparison and empirical study. And we decide to follow this preference.
>
> - (A3 & A4) Yes, you are right, we'll consider change the notion of these propositions.
>
> - (A5) Similar to the reasons given in (A1) and (A2), too little prior knowledge and the current focus of the broad zero-order optimization community (including ES and BO) lead us to weight more the practical part, as shown in Section 3.
>
> - (A6) Yes, the Fisher metric and the induced natural gradient flow are invariant, but a practical algorithm may introduce a first-order error for each step when discretizing such an invariant flow. These first-order errors can be relatively large for robust objectives such as the Discus function and neural networks, and can cumulate exponentially for typical objectives. In addition to full invariance, our novelty also consists in the first full incorporation of historical information (ref. A7) and fewer hyper-parameters compared to previous methods.
>
> - (A7) To the best of our knowledge, historical data is widely used in online optimization, but there is no general standard. Different specific settings may have different recognition on the use of historical data. For example, in convex optimization, Nesterov momentum is proven to be optimal. And in evolution strategies, all previous methods based on Gaussian do not incorporate historical data in mean update (or they claim that it is unstable to do so as in the paper of the compared method TR-CMA-ES[1]). This is a waste and we fix it.
>
> [1]Abdolmaleki, A., Price, B., Lau, N., Reis, L. P., & Neumann, G. (2017, July). Deriving and improving cma-es with information geometric trust regions. In Proceedings of the Genetic and Evolutionary Computation Conference (pp. 657-664). https://dl.acm.org/doi/abs/10.1145/3071178.3071252

---

> > ### Comment · Reviewer_wi31 · 2023-11-23
> > **reply**
> >
> > Thanks for your detailed reply and I have no further questions.

---

### Official Review · Reviewer_Rig7 · 2023-10-20

**Soundness:** 1 poor
**Presentation:** 1 poor
**Contribution:** 1 poor
**Rating:** 1
**Confidence:** 5

**Summary:**

This paper proposes an information geometric optimization (IGO) method that is invariant under all smooth bijective transformations of the model parameters.

**Strengths:**

The supplement contains running code (even it the zip file is full of superfluous stuff like MacOS and python caches).

**Weaknesses:**

Abstract: "Typical methods such as Bayesian optimization and evolutionary strategy, which stem from an online formulation that optimizes mostly through the current batch, suffer from either high computational cost or low efficiency."
I cannot accept this statement. The theoretical analysis of evolution strategies may still be under-developed. Yet, we can clearly say that their sample efficiency is close to optimal, in the sense that it matches a general lower bound on the number of samples that holds for all zeroth-order comparison based methods. I can only conclude that this paper is built on a wrong premise.

In response to the rebuttal: here is a link to a paper with a general lower runtime bound that applies to evolution strategies:
https://inria.hal.science/inria-00112820/file/lblong.pdf
It is very different from the analysis of one-max and other discrete problems.

IGO is built all around invariance principles. Therefore, calling the framework of the paper InvIGO not all but sensible. Furthermore, by design, all IGO algorithms have the desired invariance property in first order approximation. Update steps in high dimensions are necessarily small, hence higher-order differences of steps are tiny. This means that there is simply no margin for improvement over existing IGO methods. Against this background, it remains particularly unclear why the proposed method should be particularly suitable for high-dimensional problems.

I perceive section 2 as entirely chaotic. Equation (3) comes out of the blue. Following equation (5) a bit later, the paper talks about a loss function. However, we are dealing with general objectives, not with statistical data and loss functions. Assumption 1 and Definition 1 seem to be completely disconnected from the previous discussion.

The authors list "it has no external learning rate" as an advantage of their method over CMA-ES. I checked the code and saw lots of magic constants, some of which seem to be taken directly from CMA-ES. Also, I really don't know what that statement is supposed to mean, since the notion of an "external learning rate" does not exist in CMA-ES.

Experimental results: Table 1 is not acceptable. It shows a snapshot for one point in time. I want to see ECDF plots showing the time evolution of performance as they are standard in empirical research in the field (please refer to the COCO/BBOB framework for details). Figure 1 is even worse, since it fails to display function values on a logarithmic scale. Again, this violates all standard in the field, which it is unacceptable. Worse, I have to take this as a clear hint at weak understanding of evolution strategies and their linear convergence guarantees. The term "Near global optimum performance", which is crucial for reading the results, is not defined.

I tried the software provided as a supplement on the well-known Rosenbrock benchmark function in 10d:
  python exp.py --optimizer SynCMA --func rosenbrock --dim 10 --eval_num 100000 --rep 1
The result is that the proposed method fails to improve the initial solution! I tried CMA-ES as an alternative. There, solution quality reaches 8.03345e-06 after about 36000 evaluations and then jumps up to 8.49299e+27! What should I say? Please use the reference implementation of CMA-ES provided by its inventor, which NEVER does this. I can only conclude that the experiments are buggy and cannot be trusted.

**Questions:**

I don't have any questions that need clarification.

---

> ### Author Response · Authors · 2023-11-20
> **Reply to your concerns**
>
> Thank you for your detailed review! We understand your concerns from an evolutionary perspective. Indeed, our work is built on the foundation of ES. Before we address your concerns about the method and code, a statement about the motivation for our work may help us understand each other better:
>
> We agree with you on the potentially underappreciated reality of evolutionary strategy. Bayesian optimization, on the other hand, is gaining more recognition while dealing with black-box function optimization problems. To this end, we want to systematically develop methods that are competitive with state-of-the-art Bayesian optimization. All choices of criteria, such as natural axis instead of logarithmic axis and Mujoco tasks instead of COCO framework/BBOB testbeds, are made to meet the convenient setting in Bayesian optimization and the general online optimization community.
>
> Response to weaknesses:
>
> - (A1 to 'efficiency of ES')
> As far as we know, in the context of black-box optimization complexity, which serves as a lower bound, ES is optimal under some specific settings, such as convex or OneMax problems. But in a general setting, we do not find any material proving the optimality of ES. We would be really grateful if you could offer us some supporting work in the general setting!
>
> On the empirical level, however, Bayesian optimization usually outperforms ES in terms of sampling efficiency, as shown in experiments on Mujoco and other non-synthetic tasks. However, its computational complexity can grow exponentially with increasing dimension.
>
> - (A2 to 'IGO is built around invariance')
> On the one hand, a general first order error may cumulate exponentially in rugged task. On the other hand, under limited resources, a over small step size in high dimension may fail to offer competitive result. The room for a fully invariant optimizer may exist thereby.
>
> - (A3 to 'Section 2')
> Please refer first to the above statement, as Section 2 mainly serves as a link to the communities outside ES. A systematic framework for incorporating historical information synchronously and invariably for all parameters is offered, which we consider to be one of our novelties (in the content of Gaussian, there is no algorithm that stably incorporates historical information in the mean).
>
> - (A4 to 'No external learning rate')
> We denote the self-evolved term $\sigma$ as external learning rate, its initial value has to be tuned. SynCMA, on the other hand, does not have this parameter. In short, the only coefficient that has to be tuned in SynCMA is $\lambda_0$, of which the corresponding ablation study was executed in Section 4.4. In all other optimizers, coefficients that serve as the similar role as $\lambda_0$ are used as well.
>
> - (A5 to 'Experimental results')
> The full plots showing the time evolution of performance are offered in Appendix Section B due to page limit. Also, one may refer to the statement above of choosing criteria in Bayesian optimization and the general online optimization community. The 'Near Global Optimum Performance' is illustrated in the caption of Table 1 as 'achieve value better than 0.5'.
>
> - (A6 to 'Code on Rosenbrock')
> We really appreciate your detail review! First, by deleting the '*2' in line 78 of SynCMA.py and running with argument $\lambda_0 = 0$, Rosenbrock and other functions are successfully optimized to reach the optimum from 10d to 128d. We can explain these two modifications by claims in our paper : (1) In Section 4.4, we state an observation that lower dimension and simpler structure empirically calls for smaller $\lambda_0$. (2) In Appendix Section B, we state that the value of $\eta_c$ is twice of the corresponding parameter in CMA-ES as the compared CMA-ES is copied from a public github repository that uses several tricks. As you point out, these tricks may lead to an unstable behavior, we may just use the same value to eliminate the instability from aligning with the influence of these tricks.

---

### Official Review · Reviewer_PmkZ · 2023-10-31

**Soundness:** 1 poor
**Presentation:** 2 fair
**Contribution:** 1 poor
**Rating:** 5
**Confidence:** 5

**Summary:**

This paper proposes a novel variant of information geometric optimization (IGO) for black-box optimization. This paper addresses some weaknesses of the existing IGO framework: full invariance to the change of parameterization of the sampling distribution (e.g., the algorithmic behavior (slightly) changes whether the covariance matrix is parameterized by its elements or the elements of its Cholesky factor), and the historical information during the search. These limitations are tackled by casting the original objective function of the IGO framework to the minimization of the KL divergence, then introducing the additional KL regularization terms. The resulting algorithm has been instantiated with the Gaussian distribution, resulting in the proposed SynCMA. The proposed approach has been compared to several CMA-ES variants on 10 author-selected synthetic problems and two very simple control tasks.

**Strengths:**

As claimed in the paper,  a novel variant of IGO that is invariant to the choice of the parameterization of the sampling distribution and incorporates the historical information is proposed. In particular, the way of incorporating the historical information is interesting.

**Weaknesses:**

The author seem using the term ‘online optimization’ to mean black-box optimization. The term, online optimization, refers to the optimization problem having incomplete knowledge of the future. The use of line optimization in this paper is rather strange and confusing.

Though theoretically nice to have full invariance to the parameterization of the sampling distribution, its value in practice is limited. First of all, the natural gradient itself is conceptually invariant to the parameterization. In practice, it is not fully invariant as we take a finite learning rate update. However, the difference in the update between different parameterization is relatively small. The goodness of the invariance to the parameterization has not been evaluated in this paper. Moreover, already in the original IGO paper, namely IGO-ML, a variant of the IGO algorithm that is fully invariant to the change of parameterization is proposed. It hasn’t been mentioned in the paper and no comparison has been made.

Historical information is incorporated into the IGO framework in existing works (*). As far as I can see, their approach, incorporating the historical information as virtual solutions, is invariant to the choice of the parameterization of the sampling distribution as solutions are independent of the parameterization. It has not been mentioned in the paper.

(*) Youhei Akimoto, Anne Auger, and Nikolaus Hansen. 2014. Comparison-based natural gradient optimization in high dimension. In Proceedings of the 2014 Annual Conference on Genetic and Evolutionary Computation (GECCO '14). Association for Computing Machinery, New York, NY, USA, 373–380. https://doi.org/10.1145/2576768.2598258

The authors repeat an incorrect explanation of the original CMA-ES. Firstly on P7, it is stated that the covariance matrix is parameterized as \sigma \Sigma, where \sigma is a scalar and \Sigma is a PDS matrix. This is wrong. In the CMA-ES, the covariance matrix is parameterized by \sigma^2 \Sigma. The same mistake has been made in the description of the experimental setting on P15. Therefore, it is doubtful that the experimental setup is correct and fair.

Moreover, the author says that \sigma is the external learning rate that needs to be tuned. However, it is also completely wrong as \sigma is adapted during the optimization. Indeed, the adaptation of \sigma is the most important component of ES.

The authors tested the compared algorithms on unimodal and multimodal functions. Especially on multimodal functions, the initial sampling distribution, i.e., the initial mean vector and the initial covariance matrix of the Gaussian distribution, is very important and affects its resulting performance. However, as far as I can see, there is no explanation for it, and because of the above mentioned incorrect explanation of \sigma, it is rather unclear whether a fair comparison has been done. As all the test problems are shifted to have the global optimum at the origin, wrong settings would easily lead to unfair (or meaningless) comparison.

The hyperparameter setting does not seem to be fair. Though it is not explicitly written in the paper, I suppose that the authors uses the default hyper-parameters for the CMA-ES variants as they are not meant to be tuned for each problem instance. On the other hand, because the comparison are done on a single problem instances for each problem, I suspect that the parameters are tuned for these problem instances. If so, it is unfair. To avoid such a (possibly non-intended) unfair hyper-parameter setting, I strongly recommend the authors to evaluate the proposed approach on COCO framework, where 15 problem instances are generated and used for each of 24 functions.

**Questions:**

Why don’t you just use COCO framework (aka BBOB testbed) to compare algorithms?

Which implementation of the existing approaches have been used for the experiments?

---

> ### Author Response · Authors · 2023-11-20
> **Reply to your concerns**
>
> Thank you for your detailed and insightful review, which provides a second perspective on our work. For a better discussion, we would like to address the weaknesses and questions one by one.
>
> Responses to Weaknesses:
>
> - (A1 to 'Online Optimization')
> We agree with you that the paper solves a black-box optimization problem. However, to explicitly compare with Bayesian optimization and to hint the potential application to first-order online optimization problem as discussed below Eq.(5), we adapt this name.
>
> - (A2 to 'Fully Invariance')
> We put the evaluation of the goodness to be fully invariant in the comparison between TR-CMA-ES of the ill-scaled function Discus below Fig.1. We also realize from your review that IGO-ML is indeed invariant and we would mention it in the revised version. However, since IGO-ML itself is not a popular algorithm and it coincides with IGO algorithm for exponential family, the empirical comparison with CMA-ES should be sufficient.
>
> - (A3 to [1])
> Historical information is incorporated in several optimizers such as CMA-ES and VD-CMA[1], but none of them stably incorporate for mean update when parameterized with Gaussian. Moreover, VD-CMA is not invariant as it starts from the parameterization of (m, C) and relies on parameter-dependent settings such as $C = D(I + vv^T)D$.
>
> [1] Youhei Akimoto, Anne Auger, and Nikolaus Hansen. 2014. Comparison-based natural gradient optimization in high dimension. In Proceedings of the 2014 Annual Conference on Genetic and Evolutionary Computation (GECCO '14). Association for Computing Machinery, New York, NY, USA, 373–380. https://doi.org/10.1145/2576768.2598258
>
> - (A4 to '$\sigma$ or $\sigma^2$')
> We agree that we should follow the $\sigma^2$ formulation. However, since the optimizers such as CMA-ES are completely copied from public repositories, and the modifications of SynCMA only appear in the update_distribution function (except for some coefficients that are unique to SynCMA), the correctness of the experimental setup can be verified by anyone interested in it.
>
> - (A5 to 'external learning rate')
> We denote tuning $\sigma$ to its initial value, as optimizers are usually sensitive to it. SynCMA, on the other hand, does not have this parameter. In short, the only coefficient that needs to be tuned in SynCMA is $\lambda_0$, where the corresponding ablation study has been performed. All other optimizers also use coefficients that play a similar role to $\lambda_0$.
>
> - (A6 to 'Initial setting')
> It is our carelessness not to include the initial setting in the paper, although one can still convince oneself as in A4. The initial mean is 0, the initial covariance variance is $\sigma_{t = 0}I$ for all Gaussian-based optimizers, where $\sigma_{t = 0}$ is the initial value of $\sigma$.
>
> - (A7 to 'Hyperparameter setting')
> We describe the hyperparameter setting for SynCMA at the beginning of Section B of the Appendix. In short, we set all similar or shared coefficients to be the same as in CMA-ES.
>
> Responses to questions:
>
> - (Q1) : First, since we mostly aim at comparison with Bayesian optimization, criteria in online optimization such as natural axis and a threshold around 0.1 are used. Second, as we discussed in the limitation part of Section 3.2, fixed and untuned parameters $\eta_m, \eta_c, \lambda_0$ make SynCMA overshoot near the optimum. Therefore, we do not use the COCO framework for comparison.
>
> - (Q2) : As described in Section B of the Appendix, we copied existing approaches from public repositories that aim to provide competitive implementations.

---

### Official Review · Reviewer_huPQ · 2023-11-18

**Soundness:** 2 fair
**Presentation:** 1 poor
**Contribution:** 1 poor
**Rating:** 1
**Confidence:** 3

**Summary:**

The authors introduces an optimization framework relying on information
geometry, in particular natural gradient. The proposed framework allows
for real reparameterization invariance, usable in practice. Moreover, it
provides a learning rate-free setup, with some theoretical guaranties.

**Strengths:**

- Trying to tackle the practictal difficulties of IGO.
- Some code is provided, but I did not tried to run it.

I am not familiar enough with the optimization literature to comment on
the experimental part.

**Weaknesses:**

Although the text by itself is gramarly and syntactically correct,
overall the paper is difficult to follow. The transition are rather
abrupt and not a lot of explanation are given at each step.

The use of two names, InviGO and SyncCMA, is confusing as it mades
difficult to seen the boundaries between the two methods.

The justification on the absence of a learning rate is not sufficient,
or not clear enough. There are also a lot of hyper-parameters in the
algorithm.

Except in the title, there is no mention of the words "high-dimensional"

The chosen name, InvIGO (I guess the "inv" states for invariant ?), is
surprising since that it's the fundation of IGO to rely on invariance.

The complexity analysis is rather artificial and useless.

**Questions:**

I understand that InviGO is more generic than just Gaussian
distributions, but I have the feeling that the separation with SyncCMA
is quite artificial, don't you think it should be clearer to derivate
the full optimization algorithm in a straighforward way ?

At the end of Section 3, paragraph Limitation and future direction, what
is the suggested strategies to adapt the constants during the course of
the iterations ?

Could you clarify the statement in the title about the high dimension ?